# Accelerated Aging Induced by an Unhealthy High-Fat Diet: Initial Evidence for the Role of Nrf2 Deficiency and Impaired Stress Resilience in Cellular Senescence

**DOI:** 10.3390/nu16070952

**Published:** 2024-03-26

**Authors:** Priya Balasubramanian, Tamas Kiss, Rafal Gulej, Adam Nyul Toth, Stefano Tarantini, Andriy Yabluchanskiy, Zoltan Ungvari, Anna Csiszar

**Affiliations:** 1Vascular Cognitive Impairment, Neurodegeneration, and Healthy Brain Aging Program, Department of Neurosurgery, University of Oklahoma Health Sciences Center, Oklahoma City, OK 73104, USA; priya-balasubramanian@ouhsc.edu (P.B.); rafal-gulej@ouhsc.edu (R.G.); adam-nyultoth@ouhsc.edu (A.N.T.); stefano-tarantini@ouhsc.edu (S.T.); andriy-yabluchanskiy@ouhsc.edu (A.Y.); anna-csiszar@ouhsc.edu (A.C.); 2Oklahoma Center for Geroscience and Healthy Brain Aging, University of Oklahoma Health Sciences Center, Oklahoma City, OK 73104, USA; 3The Peggy and Charles Stephenson Cancer Center, University of Oklahoma Health Sciences Center, Oklahoma City, OK 73104, USA; 4Cerebrovascular and Neurocognitive Disorders Research Group, Eötvös Loránd Research Network, Semmelweis University, 1094 Budapest, Hungary; kiss.tamas1@med.semmelweis-univ.hu; 5International Training Program in Geroscience, First Department of Pediatrics, Semmelweis University, 1089 Budapest, Hungary; 6Department of Health Promotion Sciences, College of Public Health, University of Oklahoma Health Sciences Center, Oklahoma City, OK 73104, USA

**Keywords:** senescence, stress resistance, obesity, unhealthy diet, prediabetes, high-fat diet, endothelial cells, ageing

## Abstract

High-fat diets (HFDs) have pervaded modern dietary habits, characterized by their excessive saturated fat content and low nutritional value. Epidemiological studies have compellingly linked HFD consumption to obesity and the development of type 2 diabetes mellitus. Moreover, the synergistic interplay of HFD, obesity, and diabetes expedites the aging process and prematurely fosters age-related diseases. However, the underlying mechanisms driving these associations remain enigmatic. One of the most conspicuous hallmarks of aging is the accumulation of highly inflammatory senescent cells, with mounting evidence implicating increased cellular senescence in the pathogenesis of age-related diseases. Our hypothesis posits that HFD consumption amplifies senescence burden across multiple organs. To scrutinize this hypothesis, we subjected mice to a 6-month HFD regimen, assessing senescence biomarker expression in the liver, white adipose tissue, and the brain. Aging is intrinsically linked to impaired cellular stress resilience, driven by dysfunction in Nrf2-mediated cytoprotective pathways that safeguard cells against oxidative stress-induced senescence. To ascertain whether Nrf2-mediated pathways shield against senescence induction in response to HFD consumption, we explored senescence burden in a novel model of aging: Nrf2-deficient (Nrf2^+/−^) mice, emulating the aging phenotype. Our initial findings unveiled significant Nrf2 dysfunction in Nrf2^+/−^ mice, mirroring aging-related alterations. HFD led to substantial obesity, hyperglycemia, and impaired insulin sensitivity in both Nrf2^+/−^ and Nrf2^+/+^ mice. In control mice, HFD primarily heightened senescence burden in white adipose tissue, evidenced by increased *Cdkn2a* senescence biomarker expression. In Nrf2^+/−^ mice, HFD elicited a significant surge in senescence burden across the liver, white adipose tissue, and the brain. We postulate that HFD-induced augmentation of senescence burden may be a pivotal contributor to accelerated organismal aging and the premature onset of age-related diseases.

## 1. Introduction

As the global population undergoes a remarkable demographic transformation marked by a substantial increase in life expectancy, society faces unprecedented challenges associated with unhealthy aging [1,2,3,4,5,6]. The longevity revolution brings forth a pressing concern: the escalating burden of age-related diseases exacerbated by lifestyle risk factors [7,8,9,10,11,12,13,14,15,16]. These diseases, including cancer, cardiovascular disorders, and diseases of the central nervous system, pose substantial threats to the well-being of older individuals and the sustainability of healthcare systems worldwide [17,18,19,20,21].

A critical dimension of aging is the profound impact of lifestyle choices, particularly dietary habits [5,20,22,23,24,25,26], on the trajectory of aging and the risk of age-related diseases [27,28,29,30,31,32,33,34,35,36,37,38,39]. In this context, high-fat diets (HFDs) have emerged as a pervasive element of modern dietary patterns, characterized by their excessive consumption of saturated fats and relatively low nutritional value [40,41,42,43]. Epidemiological studies have compellingly established a robust association between HFD consumption, obesity, and the development of type 2 diabetes mellitus (T2DM) [44,45,46,47,48]. These findings, also supported by the results of preclinical studies, underscore the intimate connection between dietary choices and metabolic health, as well as their potential influence on the aging process [49,50,51,52,53,54,55,56]. What further compounds the challenges posed by HFDs, obesity, and diabetes is their synergistic interplay in accelerating the aging process and promoting the premature emergence of age-related diseases [57,58,59,60,61,62,63,64,65,66,67]. Despite the compelling epidemiological evidence linking these factors, the intricate mechanisms that drive these associations remain insufficiently understood [42,43,68,69,70,71,72,73].

One of the most prominent hallmarks of aging is the accumulation of highly inflammatory senescent cells [74,75,76,77,78,79,80,81,82,83,84,85]. These cells are characterized by an irreversible cell cycle arrest and the secretion of a plethora of pro-inflammatory mediators known as the senescence-associated secretory phenotype (SASP) [86,87,88]. The emergence of cellular senescence is a multifaceted process, influenced by factors such as oxidative stress-mediated DNA damage [89,90] and mitochondrial damage [91,92]. Increasing evidence supports the pivotal role of cellular senescence in the pathogenesis of a wide range of age-related diseases, including cancer, cardiovascular disorders, vascular cognitive impairment (VCI), and neurodegenerative conditions [93,94,95,96,97,98], primarily through the promotion of chronic low-grade inflammation, aptly termed “inflammaging” [99,100,101]. Thus, investigating the extent to which HFD consumption amplifies senescence burden across multiple organs becomes crucial in elucidating the mechanisms driving accelerated, unhealthy aging and the development of age-related diseases [102,103,104,105,106,107,108,109,110,111,112].

Impaired stress resilience, the incapacity to restore homeostasis following exposure to stressful conditions, is a hallmark of aging and plays a pivotal role in the pathogenesis of various age-related diseases [113,114,115]. A critical and evolutionarily conserved cellular mechanism responsible for maintaining redox homeostasis is the Nuclear factor-erythroid-2-related factor (Nrf2)-antioxidant response element (ARE) signaling pathway [116]. Nrf2 is a transcription factor that orchestrates the coordinated expression of numerous antioxidant and DNA repair genes, conferring cytoprotective effects against oxidative damage during periods of stress [117]. Significantly, aging is associated with Nrf2 dysfunction [117,118]. Furthermore, our previous research has demonstrated that the adverse consequences of high-fat diet (HFD) consumption, including impaired endothelial function, cerebral blood flow regulation, blood-brain barrier disruption, neuroinflammation, and cognitive dysfunction, are exacerbated in mouse models with Nrf2 deficiency, effectively mimicking the aging phenotype [59,119]. Nrf2 knockout mice fed a HFD exhibit elevated levels of SASP factors such as IL-1β and TNFα, compared to their wild-type counterparts [119]. Against this backdrop, we postulate that Nrf2 deficiency may accelerate aging by exacerbating cellular senescence.

The present study was designed to investigate whether HFD consumption, leading to obesity and type 2 diabetes mellitus, amplifies senescence burden across multiple organs. Additionally, we hypothesized that age-related Nrf2 dysfunction further magnifies the deleterious effects of HFD on senescence. While Nrf2 knockout mice (Nrf2^−/−^) have yielded invaluable insights into the protective role of Nrf2 in age-related diseases, they do not entirely replicate physiological aging, where there is a partial rather than complete loss of Nrf2 activity. Thus, in this study, we conducted experiments using heterozygous Nrf2^+/−^ mice, characterized by partial Nrf2 activity loss. These mice, along with their wildtype littermates, were subjected to a 6-month HFD regimen, and subsequently, senescence burden in the liver, white adipose tissue, and the brain was assessed.

## 2. Materials and Methods

### 2.1. Animals and Treatment

To elucidate the interplay between metabolic stress, impaired stress resilience and senescence in a mouse model of accelerated aging with partial Nrf2 deficiency, we created heterozygous Nrf2 knock-out (Nrf2^+/−^) mice on a p16-3MR background. The p16-3MR senescence reporter mouse model was developed by Dr. Judith Campisi at the Buck Institute on Aging. These mice express the 3MR transgene construct containing luciferase, red fluorescent protein, and thymidine kinase under the control of p16 promoter [120,121,122,123,124,125,126,127,128]. Crossbreeding was conducted between p16-3MR mice and Nrf2 homozygous knock-out mice (Jackson Labs, Farmington, CT, B6.129X1-Nfe2l2tm1Ywk/J) to generate the Nrf2^+/−^ mice on the p16-, 3MR background. For genotyping, DNA was extracted from ear punches and PCR was performed using the Red Extract-N-Amp PCR kit (Sigma, St. Louis, MO, USA). The primers used for the PCR include: GCC TGA GAG CTG TAG GCC C (common forward primer), GGA ATG GAA AAT AGC TCC TGC C (WT reverse) and GAC AGT ATC GGC CTC AGG AA (mutant reverse) (representative genotyping figure differentiating the homozygous and heterozygous Nrf2 mice is provided in Appendix A). Wildtype littermates served as controls (Nrf2^+/+^ mice).

Between 4 to 6 months of age, both Nrf2^+/+^ and Nrf2^+/−^ mice (15 males and 14 females, total *n* = 29) were randomly assigned to one of two dietary groups: a standard chow diet (SD, containing 10% calories from fat) or a high-fat diet (HFD, consisting of 60% calories from fat; Research Diets Inc. New Brunswick, NJ, D12450B). The dietary regimen continued for a duration of 6 months, with the mice having ad-libitum access to both food and water. Throughout the experimental period, bi-weekly body weight measurements were recorded. Upon completion of the 6-month treatment period, all animals underwent perfusion with ice-cold PBS. Subsequently, brain, epididymal fat, and liver tissues were collected, rapidly frozen in liquid nitrogen, and preserved at −80 °C until further analysis. The animal protocols adhered to the ethical guidelines and were approved by the Institutional Animal Care and Use Committee (Protocol number#21-030, approval date—5 December 2021) at the University of Oklahoma Health Sciences Center.

### 2.2. Fasting Glucose and Insulin Measurements

During the fifth month of dietary intervention, we collected blood samples from overnight-fasted animals to measure glucose levels using the OneTouch Ultra Blue glucometer and insulin levels using the Ultra-Sensitive Mouse Insulin ELISA Kit from CystalChem, Elk Grove Village, IL. To assess insulin resistance, we calculated the Homeostasis Model Assessment of Insulin Resistance (HOMA-IR) index using the formula [fasting serum glucose × fasting serum insulin/22.5].

### 2.3. Nrf2 Activity Assay

Nuclear proteins were isolated from frozen liver and cortex samples using the Nuclear Extraction Kit from Active Motif, Carlsbad, CA, USA. We quantified the protein concentrations using the Micro BCA assay from Thermo Fisher Scientific, Rockford, IL, USA. Equal concentrations of isolated nuclear protein (30 µg) were employed to assess Nrf2 DNA-binding capability using the TransAM Nrf2 transcription factor assay kit from Active Motif. In brief, nuclear protein extracts were added to a 96-well plate coated with antioxidant-response element (ARE) sequence oligos and incubated at room temperature for 1 h. After washes to remove unbound proteins, anti-Nrf2 antibodies (1:1000) were added to the wells and incubated for 1 h. Following further washes, HRP-conjugated secondary antibodies were added and incubated for another hour. After additional washes, a substrate solution was added for color development, and optical density (OD) values were measured at 450 nm using a Tecan Spark multimode microplate reader.

### 2.4. In Vivo Bioluminescence Measurements 

Mice were injected with 15 μg of Xenolight RediJect Coelentarazine h intraperitoneally. After 25 min, the mice were anesthetized with isofluorane, and luminescence was measured with a Xenogen IVIS-200 Optical in vivo imaging System (5 min exposure time). 

### 2.5. Immunohistochemistry

Brains collected after perfusion with ice-cold PBS were fixed in 4% PFA for 24 h and subsequently cryoprotected through immersion in 30% sucrose, followed by embedding in OCT medium. Frozen brain OCT blocks were cryosectioned at a thickness of 35 µm, and the sections were stored in cryoprotectant solution at −20 °C until staining. The free-floating sections were immunolabeled using an anti-endomucin primary antibody (1:75, EMD Millipore, Burlington, MA, MAB2624) and an anti-rat Alexa 488 secondary antibody (1:500, Invitrogen, Molecular Probes, Carlsbad, CA, USA) to identify capillary endothelial cells in the mouse brain. Nuclear counterstaining was performed using DAPI (Sigma). Confocal images were acquired using a Leica SP8 MP confocal laser scanning microscope. Senescent endothelial cells were identified in 20× images by co-localizing RFP (p16+ senescent cells) with Alexa 488 green fluorescence signal in the hippocampal region. Double-positive cells were quantified using the Image J count plugin and expressed as the average number of senescent endothelial cells per field in each group.

### 2.6. Real-Time PCR

RNA extraction from frozen tissue samples was carried out using the RNeasy Miniprep kit from Qiagen, Germantown, MD, USA. Equal amounts of RNA (1 µg) were reverse transcribed to cDNA using the High-Capacity RNA to cDNA kit from Applied Biosystems, San Francisco, CA, USA. Real-time PCR reactions were performed in a QuantStudio 12K Flex Real-Time PCR System from Applied Biosystems using validated TaqMan probes for p16, Nrf2, and the housekeeping gene, β-actin (Applied Biosystems), with 50 ng cDNA per reaction as previously reported [119,129]. Data were analyzed using the 2^ΔΔCT^ method, where CT represents the threshold cycle.

### 2.7. Statistical Analysis

All data are presented as mean ± SEM. Statistical analyses were performed using one-way ANOVA followed by Bonferroni’s post hoc test or Student’s *t*-test as appropriate. A *p*-value of <0.05 was considered statistically significant.

## 3. Results

### 3.1. Partial Nrf2 Ablation in Nrf2^+/−^ Mice and Its Impact on Nrf2 Activity

To assess whether Nrf2^+/−^ mice exhibited a physiological age-related reduction in Nrf2 rather than complete ablation, we examined Nrf2 transcript levels and activity in both the brain cortex and liver tissues. As anticipated, Nrf2^+/−^ mice displayed an approximate 50% reduction in Nrf2 mRNA levels in the cortex (Figure 1A). Similarly, we observed a significant decrease in Nrf2 activity, as indicated by DNA binding ability in nuclear extracts, in the cortex and a nearly significant reduction in Nrf2 activity in the liver (Figure 1B,C).

### 3.2. Metabolic Parameters in Nrf2^+/−^ Mice and the Effects of HFD

Next, we investigated the impact of partial Nrf2 ablation and HFD on metabolic parameters. Both Nrf2^+/+^ and Nrf2^+/−^ mice exhibited comparable increases in final body weight and fasting glucose levels when compared with their SD controls (Figure 2A,B). HFD treatment also induced similar increases in fasting insulin levels and HOMA-IR, a measure of insulin resistance, in both groups, although it reached statistical significance only in the Nrf2^+/+^ group when compared to their SD controls (Figure 2C,D).

### 3.3. Impact of Partial Nrf2 Ablation on Senescence Induction in Various Tissues

Previously, we demonstrated that both aging and HFD-induced oxidative stress promoted the induction of p16-mediated senescence in Nrf2 knock-out mice (complete ablation of Nrf2 gene and activity) [119,129]. In this study, we aimed to investigate whether partial Nrf2 ablation, which mimics physiological aging, could recapitulate the effects of HFD-induced metabolic stress on senescence. First, we measured luminescence signals from renilla luciferase expressed in p16+ cells as a surrogate marker for senescence in vivo. HFD treatment significantly increased whole-body luminescence in Nrf2^+/−^ mice compared to Nrf2^+/+^ group (Appendix A). Next, we assessed the expression levels of the well-established senescence marker, cyclin-dependent kinase inhibitor p16INK4a (*Cdkn2a*), in multiple tissues.

In the liver, HFD did tended to increase *Cdkn2a* expression in the Nrf2^+/+^ group, but this change did not reach statistical significance. In contrast, we observed a several-fold increase in p16 mRNA levels in Nrf2^+/−^ mice on HFD compared to SD-fed Nrf2^+/−^ mice and both Nrf2^+/+^ groups (Figure 3A), suggesting a significant interaction between diet and partial Nrf2 deficiency in accelerating hepatic senescence. In the visceral adipose tissue (eWAT), HFD significantly increased *Cdkn2a* expression both in Nrf2^+/+^ and Nrf2^+/−^ mice, with Nrf2 deficiency not exerting an additive effect (Figure 3B). 

Given that senescent cells constitute a rare population in the brain (comprising 1–2% of total brain cells), we employed mRFP as a biomarker to identify and quantify p16+ senescent cells in the brain, focusing on senescent endothelial cells in particular, as recent single-cell sequencing studies indicate their heightened susceptibility to senescence in the aging brain [130]. The analysis of p16^+^/endomucin^+^ cells revealed that in the hippocampi of Nrf2^+/+^ mice, a significant increase in the number of senescent endothelial cells was not observed following HFD. Conversely, in the hippocampi of Nrf2^+/−^ mice, HFD resulted in a noteworthy increase in the population of senescent endothelial cells (Figure 4A,B). Overall, our findings suggest that partial Nrf2 ablation was sufficient to accelerate metabolic stress-induced senescence in multiple organs, mimicking the aging phenotype.

## 4. Discussion

In this study, we delved into the intricate relationship between HFDs, partial Nrf2 deficiency, and cellular senescence to decipher their collective impact on the process of aging. Our investigation yielded several important initial findings that collectively contribute to our understanding of the complex mechanisms governing senescence and unhealthy aging. We observed that partial Nrf2 ablation in Nrf2^+/−^ mice closely resembled the physiological age-related reduction in Nrf2, indicating a relevant model for exploring the influence of moderate Nrf2 deficiency on senescence induction. Additionally, both Nrf2^+/+^ and Nrf2^+/−^ mice exhibited similar metabolic responses to HFD, including increases in body weight and fasting glucose levels, with a tendency toward insulin resistance. Notably, the effects of HFD and Nrf2 deficiency on senescence markers varied across tissues, highlighting the intricate interplay of factors in different physiological contexts. Importantly, in the brain, we uncovered a significant increase in senescent endothelial cells in Nrf2^+/−^ mice exposed to HFD, a finding with potential implications for age-related cognitive decline. 

Aging is a multifaceted process influenced by a dynamic interplay of genetic, environmental, and lifestyle factors, all of which collectively contribute to the emergence of age-related diseases. Among these factors, dietary choices, notably the consumption of high-fat diets (HFDs), have emerged as a focal point of scientific inquiry due to their pronounced association with conditions such as obesity, diabetes, accelerated aging, and heightened susceptibility to age-related diseases, leading to increased morbidity and mortality [51,55,69,72,115,131,132,133,134,135,136]. In the context of unhealthy aging, our study endeavors to illuminate the intricate relationship between HFDs, Nrf2 deficiency, and the phenomenon of cellular senescence.

The age-related decline in Nrf2 or its homolog levels and activity has been well-documented across various tissue types [137,138,139,140] and in diverse species, spanning from invertebrates to humans [140,141,142,143]. It can be attributed to a convergence of factors that collectively contribute to Nrf2 dysfunction and impaired stress resilience. Under normal physiological conditions, Nrf2 is kept at low levels in the cytoplasm through interaction with the E3 ubiquitin ligase complex, primarily composed of Kelch-like ECH-associated protein (KEAP1). This complex plays a pivotal role in negatively regulating Nrf2 by facilitating its ubiquitination and subsequent proteasomal degradation. However, increased oxidative stress induces conformational changes in the Nrf2-KEAP1 complex prevent ubiquitination, leading to the accumulation of Nrf2 and its subsequent translocation to the nucleus. In the nucleus, Nrf2 forms heterodimers with small MAF (sMAF) proteins and binds to antioxidant response element (ARE) sequences in gene promoter regions, initiating the transcription of crucial cytoprotective genes [144]. In aging, the dysregulation of this finely tuned system, driven by factors such as alterations in KEAP1-Nrf2 interactions, and the diminished availability of critical cofactors, ultimately contributes to Nrf2 dysfunction and impaired stress resilience [119,129,138,145]. These intricate molecular changes collectively underlie the age-related decline in Nrf2 activity, rendering cells and tissues more vulnerable to the detrimental effects of oxidative stresses. Our results demonstrate that partial Nrf2 ablation in Nrf2^+/−^ mice closely mimics the physiological age-related reduction in Nrf2, evident by a significant decrease in Nrf2 mRNA levels and Nrf2 activity in both the cortex and liver tissues. This partial loss of Nrf2 function in Nrf2^+/−^ mice allowed us to explore the impact of moderate Nrf2 deficiency, which is more representative of physiological aging, on senescence induction.

Impairment in Nrf2 signaling during the aging process leads to maladaptive stress responses and stands as a pivotal contributor to the pathogenesis of age-related diseases [119,138,146,147]. Metabolic stress is a well-established factor in promoting cellular senescence [102,148,149]. Our studies underscore the significant role of Nrf2 in preserving redox homeostasis [138,141] and mitigating cellular senescence under conditions of HFD-induced metabolic stress, which closely resemble the conditions of unhealthy aging in humans. Specifically, we observed that partial Nrf2 deficiency (Nrf2^+/−^) was adequate to exacerbate HFD-induced senescence within cerebromicrovascular endothelial cells situated in the hippocampus—a brain region vital for the regulation of learning and memory function. These findings expand upon our prior research, where complete Nrf2 ablation (Nrf2^−/−^) induced senescence in cerebral arteries, hastening cerebromicrovascular dysfunction and cognitive decline in HFD-fed mice [119]. Furthermore, in addition to metabolic stressors, complete absence of Nrf2 signaling (Nrf2^−/−^) amplified the impact of age-related oxidative stress on senescence and inflammation within the cerebral vasculature [129]. Taken together, these collective findings underscore the protective role of Nrf2 signaling in mitigating senescence among cerebromicrovascular endothelial cells, particularly in the context of unhealthy brain aging and obesity-related metabolic stressors. 

Dysfunction in Nrf2 has been intricately linked to the development and progression of age-related diseases, including the pathogenesis of VCI [59,119,129,138,145,150]. Notably, previous investigations conducted by our research group have provided compelling evidence that Nrf2 deficiency exacerbates metabolic stress-induced cerebromicrovascular dysfunction [59,119,150]. This dysfunction is characterized by heightened blood-brain barrier permeability, neurovascular uncoupling, and neuroinflammation [59,119]. It is plausible that this accelerated aging phenotype observed in the brain due to Nrf2 deficiency can be mechanistically attributed to the induction of senescence under metabolic stress conditions within the cerebral microcirculation [119,129]. 

Supporting this hypothesis, substantial evidence indicates that senescence exerts detrimental effects on endothelial-dependent dilation by diminishing endothelial nitric oxide synthase (eNOS) activity [95,151], impairs angiogenesis [138], reduces the expression of tight junction proteins [152], and increases barrier permeability [153]. Furthermore, in addition to directly promoting tissue dysfunction and inflammation, senescent cells also profoundly influence the function and phenotype of neighboring cells through the paracrine secretion of SASP factors, which amplify the effects of a small population of senescent cells. Notably, robust evidence suggests that the elimination of senescent cells from the brain, utilizing various senolytic strategies, can effectively restore blood-brain barrier integrity, enhance the regulation of cerebral blood flow, and augment brain capillarization in a range of aging and accelerated brain aging models [95,97,121,154]. These improvements are closely correlated with heightened cognitive performance. Importantly, these models encompass not only mouse models of aging but also instances of cognitive decline induced by whole-brain irradiation and chemotherapy [97,121]. Therefore, further investigations are warranted to rigorously evaluate the efficacy of senolytic treatments within the context of HFD-induced accelerated brain senescence, representing a critical area that merits comprehensive exploration.

An intriguing finding of our study pertains to the differential impact of HFD and Nrf2 deficiency on senescence burden in various tissues. Notably, in the liver, HFD did not significantly elevate senescence burden in Nrf2^+/+^ mice. However, in Nrf2^+/−^ mice, we observed a substantial increase in senescence burden upon exposure to HFD, suggesting a robust interaction between diet and Nrf2 deficiency in driving hepatic senescence. Furthermore, in the context of visceral adipose tissue, HFD induced senescence regardless of genotype, with no additional impact of Nrf2 deficiency. These tissue-specific variations underscore the complexity of senescence regulation influenced by distinct factors and emphasize the necessity for further investigations into their specific molecular mechanisms. Importantly, understanding the implications of liver and adipose senescence for brain aging phenotypes is essential and warrants focused exploration in future research endeavors. Moreover, it is noteworthy that SASP factors secreted by senescent cells [84,155] within the liver and adipose tissue are likely to exert significant influences on BBB integrity [156,157,158,159] and cerebromicrovascular endothelial function and phenotype [160,161,162,163], further accentuating the intricate interplay between these organs in the context of unhealthy aging.

## 5. Conclusions

In summary, our study aligns with the conclusions of previous research, highlighting the detrimental effects of HFDs on the aging process. Our study provides valuable insights into the complex web of interactions between HFDs, Nrf2 deficiency, and cellular senescence in the context of unhealthy aging (Figure 5). These findings enhance our understanding of the multifaceted mechanisms governing senescence and the potential implications for age-related diseases. Our research has underscored the pivotal role of Nrf2 in alleviating senescence, particularly under conditions of metabolic stress affecting the cerebral microcirculation. This emphasizes the potential significance of Nrf2 in mitigating age-related cognitive decline. To unravel the complexities of the involved molecular pathways and to explore potential therapeutic interventions targeting the Nrf2-ARE pathway for ameliorating the detrimental consequences of unhealthy aging, further in-depth investigations are warranted.

As we delve deeper into the intricacies of these molecular pathways, it becomes increasingly imperative to translate these discoveries into practical strategies for promoting healthy aging. Aligning with well-established knowledge, the pivotal role of dietary choices in fostering healthy aging warrants heightened attention. Experts in the field strongly advocate for the maintenance of a well-balanced and nutritionally rich diet, one replete with antioxidants and anti-inflammatory foods, which play a fundamental role in supporting Nrf2 activity while counteracting oxidative stress and inflammation [72,164]. Specifically, consideration should be given to phytochemicals such as sulforaphane from broccoli, curcumin from turmeric, and resveratrol from grapes which have shown promise in activating Nrf2 and exerting beneficial effects on aging-related processes. Some of the putative mechanisms behind the anti-aging effects of these phytochemicals include increased anti-oxidant and anti-inflammatory effects, improved mitochondrial function and improved autophagy and proteostasis [165,166]. Findings from clinical studies also support this by demonstrating a positive association between higher phytochemical intake with lowered risk for cardiometabolic and neurodegenerative diseases [167,168]. Such dietary choices are pivotal in mitigating metabolic stress and the associated risks that can accelerate senescence. To fully harness the potential of these insights, it is imperative that we invest in further research and clinical studies aimed at evaluating the effectiveness of dietary interventions designed to enhance Nrf2-mediated cellular resilience and mitigate the onset of cellular senescence. In doing so, we can aim to transform scientific knowledge into practical, evidence-based strategies firmly grounded in geroscience. These strategies empower individuals to embrace and embark on the path of healthy aging. 

## Figures and Tables

**Figure 1 nutrients-16-00952-f001:**
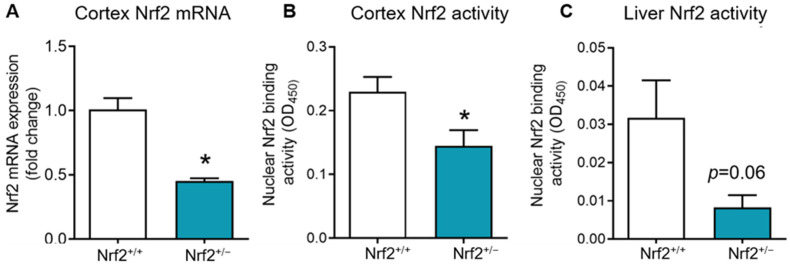
Nrf2 gene expression and DNA binding activity in Nrf2^+/+^ and Nrf2^+/−^ mice. (**A**): Nrf2 mRNA levels in the cortex of Nrf2^+/+^ and Nrf2^+/−^ mice (*n* = 3–5). (**B**,**C**): DNA binding ability of Nrf2 assessed in the nuclear extracts of the cortex and liver of Nrf2^+/+^ (*n* = 4) and Nrf2^+/−^ (*n* = 4–5) mice using Trans−AM motif assay. Data are presented as mean ± SEM. * indicate a significant difference (*p* < 0.05) from Nrf2^+/+^ mice.

**Figure 2 nutrients-16-00952-f002:**
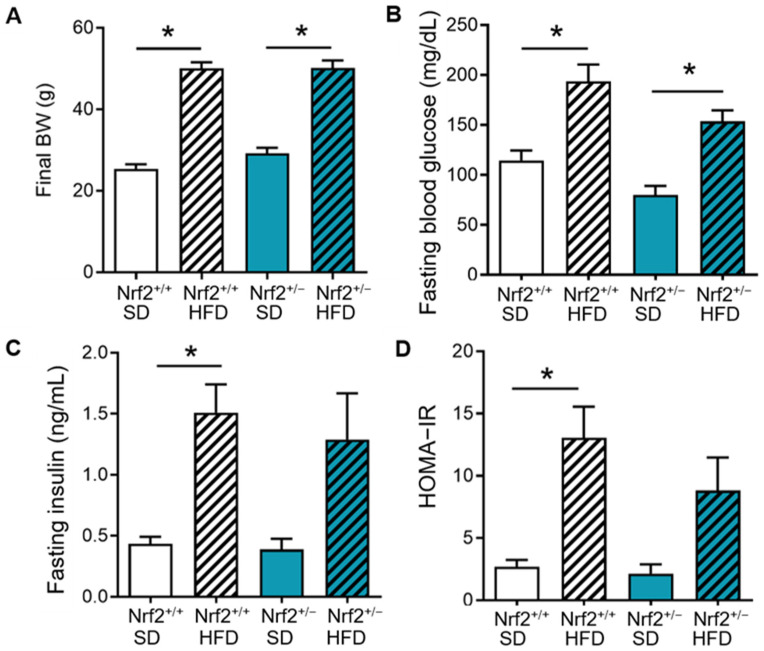
Impact of partial Nrf2 ablation and HFD on metabolic parameters. Nrf2^+/+^ and Nrf2^+/−^ mice were fed with standard diet (SD) or high-fat diet (HFD) for 20 weeks. (**A**–**D**): Final body weight (BW), fasting blood glucose, fasting insulin, and HOMA-IR index measured in Nrf2^+/+^ (*n* = 7–9) and Nrf2^+/−^ (*n* = 6–7) mice after 20 weeks of respective dietary treatment. Data are presented as mean ± SEM. * denotes a significant difference (*p* < 0.05) from the indicated group.

**Figure 3 nutrients-16-00952-f003:**
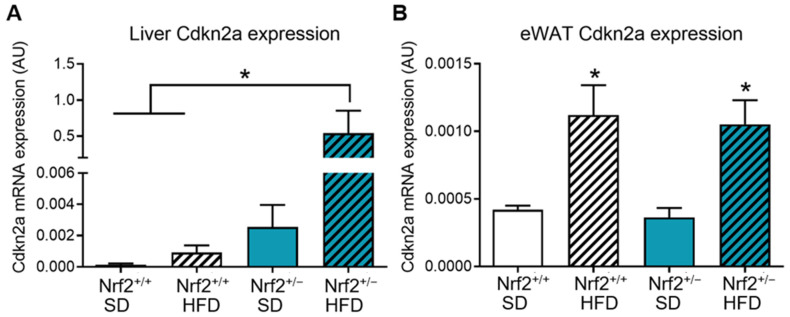
Partial Nrf2 ablation increases the expression of senescence markers in peripheral tissues. Analysis of *Cdkn2a* (p16^INK4a^) gene expression in the liver (**A**) and epididymal white adipose tissue (eWAT) (**B**) samples from Nrf2^+/+^ and Nrf2^+/−^ mice after 20 weeks of consuming a standard diet (SD) or high fat diet (HFD) (*n* = 3−6/group). Data are presented as mean ± SEM. * indicates a significant difference (*p* < 0.05) from the indicated group.

**Figure 4 nutrients-16-00952-f004:**
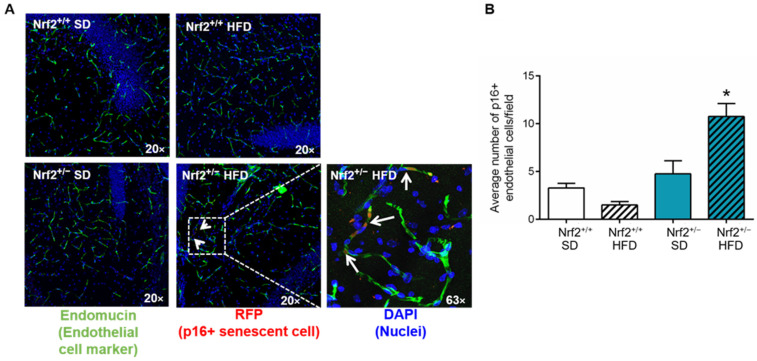
Induction of senescence in brain microvascular endothelial cells due to partial Nrf2 ablation. (**A**) Representative images of immunofluorescent labeling used to detect p16-RFP in endomucin+ endothelial cells in the hippocampus of Nrf2^+/+^ and Nrf2^+/−^ mice after 20 weeks of consuming a standard diet (SD) or high fat diet (HFD) (*n* = 4/group). White arrows indicate the presence of mRFP positive senescent endothelial cells in the hippocampus. (**B**) Quantification of the p16-RFP+/endomucin+ endothelial cells in the hippocampi of all the groups. Data are presented as mean ± SEM. * *p* < 0.05.

**Figure 5 nutrients-16-00952-f005:**
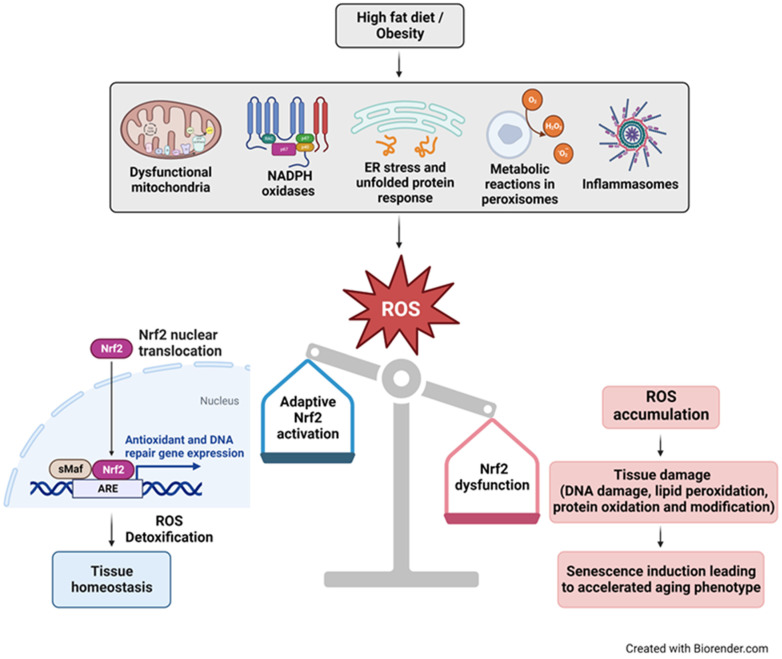
Schematic illustration of the relationship between consumption of HFD/obesity, oxidative stress, Nrf2 deficiency, and cellular senescence in accelerated aging. Dysfunctional mitochondria, NADPH oxidases, ER stress from misfolded proteins, metabolic processes in peroxisomes, and inflammasomes are major sources of increased reactive oxygen species (ROS) levels in obesity. Nrf2 plays a crucial role in maintaining cellular redox homeostasis. Under conditions of oxidative stress, Nrf2 is released from the Keap1−Cul3−RBX1 complex, after which it translocates into the nucleus, forms heterodimers with small Maf proteins (sMaf), and binds to antioxidant response elements (AREs). This activation leads to the transcription of genes encoding antioxidant enzymes, DNA repair mechanisms and proteins that confer anti-inflammatory effects, facilitating ROS detoxification and restoration of cellular homeostasis. Conversely, impaired Nrf2 activation results in the accumulation of toxic levels of ROS, inducing macromolecular damage like DNA breaks, and triggering senescence, ultimately leading to an accelerated aging phenotype.

## Data Availability

Data for further analysis will be available to interested researchers upon direct request to the corresponding author.

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
