# Peer review of "Accelerated Aging Induced by an Unhealthy High-Fat Diet: Initial Evidence for the Role of Nrf2 Deficiency and Impaired Stress Resilience in Cellular Senescence"

_nutrients, 2024, doi:10.3390/nu16070952_

Round 1

Reviewer 1 Report

Comments and Suggestions for Authors

The article presented by Balasubramanian and collaborators attempts to clarify the role of Nrf2 in accelerated tissue aging in animals fed a high-fat diet. For this reviewer it is an interesting article, but incomplete. The model and experimental design seem correct to respond to the objective of the study. However, a heterozygous mouse between p163MR and Nrf2-/- is developed to obtain the Nrf2+/- heterozygous mouse having p163MR as a genetic background. It is an excellent model, but no history is presented to confirm that they are working with these animals. It would be very good for this reviewer to be able to observe the results of the genotyping of the animals in some complementary figure. On the other hand, the title states that the high-fat diet they used induces accelerated aging, but no antecedents are shown that can validate this model as a plausible model of aging. Do you have data from this same model, but using aged animals?

Regarding the results of qPCR (p16) to evaluate induction of senescence in tissues, it is a widely used method, but the gold standard to evaluate senescence is beta galactosidase staining. It would be convenient to have beta gal staining in the analyzed tissues or alternatively an image showing the fluorescence of the 3MR transgene, this could help the reviewer understand the level of senescence of each tissue, the expression of p16 alone is not an acceptable parameter.

Finally, what worries me most is that their conclusions are based more on the available literature than on the results, the experiments show that HFD increases cellular senescence differentially in the tissues, they quickly focus on the brain and propose that the senescence observed in the microvasculature of the hippocampus could be the cause of cognitive deterioration, but there are no results that evidence this. No evidence of behavior or learning is shown, nor are systemic parameters shown that could correlate with this. For this reviewer, it is necessary to add some experiment (permeability assays, nitric oxide production or expression and activation of eNOS) that allow evaluating in vivo or in vitro the function of brain endothelial cells in Nrf2+/- mice.

Reviewer 2 Report

Comments and Suggestions for Authors

Review

The authors have investigated the combined effect of a high-fat diet (HFD) and NrF2 deficiency by giving HFD to Nrf2+/- (p16-3MR background) mice.

HFD and Nrf2+/- phenotypes have been previously studied but this paper addresses the combined impact with a focus on cellular senescence for the first time.

The impact of these results is relatively low, as it is more than expected that reducing antioxidant response during a high-fat diet may be deleterious. It would be instead much more interesting to see if improving Nrf2 response can protect against a high-fat diet.

Anyway, there is some novel information that may eventually deserve publication. However, the results are still not enough to support the conclusions and the whole manuscript seems more a preliminary study rather than a well-planned series of experiments.

In my opinion, this manuscript has to be ameliorated in several aspects and some experiments need to be added before publication.

In details:

The authors need to check and revise various parts of the manuscript:

1)      Please specify the n. of approval of the ethical protocol in the methods.

2)      Figure 1: why mRNA in the liver is not reported. Anyway, this figure states that Nrf2 is impaired only in the cortex and not in the liver.

3)      The sex of the mice is not reported, please check and revise. Manipulation of Nrf2 may have sex-specific effects.

4)      There are some comparisons with n = 3. I’m not sure that this is enough. It was a sample size planned before starting the study? What is the effect size?

5)      Figure 2:

a)       Panel B. It seems that Nrf2+/+ mice have higher fasting glucose than Nrf2+/-. Is this significant?

b)      Panel C and D. Why Nrf2+/- do not have increased fasting insulin? Is this only a sample size problem? If this is the case, please increase the number of mice, otherwise please try to understand why these mice have impaired insulin secretion under HFD.

6)      “….our findings suggest that partial Nrf2 ablation was sufficient to accelerate metabolic stress…….mimicking the aging phenotype”. Please show the aging phenotype in Nrf2+/+ mice .

7)      Figure 4:

a)       Panel A, RFP positive senescent cells (I think marked in red) are not visible in the mid panels. Please check.

b)      It seems that p16+ cells are lowered in HFD Nrf2+/+ mice, is this significant, please check and eventually explain.

c)       Is p16 the sole marker used for senescence? Albeit these are transgenic mice expressing RFP in p16+ cells I would add further staining to confirm the senescence phenotype and avoid any potential artificat (considering also the relatively low sample size). The basal number of senescent cells seems excessive (around 3-4% in 12-month-old wild-type mice), so I believe that this analysis needs further investigation.

8)      Why quantification of p16+ cells in liver and fat was not performed by RFP?

9)      The authors may improve the last part of the discussion around Geroscience by considering phytochemicals that may improve Nrf2.

Round 2

Reviewer 1 Report

Comments and Suggestions for Authors

Dear authors, Thank you for responding and explaining each of the points I had doubts about. The article improved significantly.

I take this opportunity to tell you that I was not able to see the supplementary figures of the new version, I would have liked to see the supplementary figures that responded to my comments.

Author Response

We apologize that the supplementary figures document was not accessible before. Please see attached the file with the supplementary figures. We also included the DOI link in the manuscript submission. Thank you.

Reviewer 2 Report

Comments and Suggestions for Authors

The authors have partly addressed the concerns. Some requests have not been completely addressed, but I understand that adding experiments with mice is now a  challenge due to ethical, costs and time problems.

So, as stated by the authors these should be considered as preliminary experiments, and this should be clearly reported in the Title, abstract and discussion.

Importantly, the authors have indicated now that the experiments have been performed in both sexes, but it is important to state "how many males and females" for each experiments in the figure legends.

In my opinion, these few additional things are fixed.

Author Response

We understand the reviewer's concern. 

We have made it clear in the title, abstract and discussion. The title has been changed to 

Accelerated Aging Induced by an Unhealthy High-Fat Diet: Initial Evidence for the Role of Nrf2 Deficiency and Impaired Stress Resilience in Cellular Senescence

We have also updated the number of males and females in the methods section under animals and treatment. Unfortunately, we could not have an accurate count of males and females for each figure legend. 

Thank you